# Measuring the distribution of fitness effects in somatic evolution by combining clonal dynamics with dN/dS ratios

Marc J Williams[1,2]*, Luis Zapata[3], Benjamin Werner[1], Chris P Barnes[4], Andrea Sottoriva[3]*, Trevor A Graham[1]*

[1]Centre for Genomics and Computational Biology, Barts Cancer Institute, Barts and the London School of Medicine and Dentistry, Queen Mary University of London, London, United Kingdom; [2]Computational Oncology, Department of Epidemiology and Biostatistics, Memorial Sloan Kettering Cancer Center, New York, United States; [3]Evolutionary Genomics and Modelling Lab, Centre for Evolution and Cancer, The Institute of Cancer Research, London, United Kingdom; [4]Department of Cell and Developmental Biology, University College London, London, United Kingdom

**Abstract** The distribution of fitness effects (DFE) defines how new mutations spread through an evolving population. The ratio of non-synonymous to synonymous mutations (dN/dS) has become a popular method to detect selection in somatic cells. However the link, in somatic evolution, between dN/dS values and fitness coefficients is missing. Here we present a quantitative model of somatic evolutionary dynamics that determines the selective coefficients of individual driver mutations from dN/dS estimates. We then measure the DFE for somatic mutant clones in ostensibly normal oesophagus and skin. We reveal a broad distribution of fitness effects, with the largest fitness increases found for TP53 and NOTCH1 mutants (proliferative bias 1–5%). This study provides the theoretical link between dN/dS values and selective coefficients in somatic evolution, and measures the DFE of mutations in human tissues.

**\*For correspondence:**
william1@mskcc.org (MJW);
andrea.sottoriva@icr.ac.uk (AS);
t.graham@qmul.ac.uk (TAG)

**Competing interests:** The authors declare that no competing interests exist.

## Introduction

One of the principal goals of large-scale genome sequencing of somatic tissues is to uncover genetic loci under positive selection, so-called 'driver' genes, that lead to clonal expansions. Measurement of the selective advantage of each driver mutation enables prediction of future evolutionary dynamics (*Williams et al., 2019*), provided the selective regime remains constant. In evolutionary biology, the distribution of fitness effects (DFE) is a fundamental entity that describes the selective consequences of a (large) number of individual mutations of an ancestral genome (*Eyre-Walker and Keightley, 2007*). In somatic evolution, particularly in cancer genomes, we have an extensive knowledge of the catalogue of recurrent, and likely positively selected, somatic mutations (*Martincorena et al., 2017*), but the fitness changes associated with each mutation remain largely unquantified.

Extensive experimental effort is ongoing to determine the fitness effects of mutations. Most prominently is lineage tracing of mutations in mouse models (*Vermeulen et al., 2013*; *Rogers et al., 2018*), but these methods are not sufficiently high-throughput to produce the DFE for all somatic mutations. Other studies have estimated the selective coefficient of somatic mutations by measuring their frequency over time in the same individual using longitudinal sampling (*Körber et al., 2019*), however this method is broadly limited to somatic evolution in the blood (*Gibson and Steensma,*

*2018*) (where it is feasible to take samples from healthy individuals over time) and in rare cases of patients under active surveillance.

An alternative approach is to infer selective coefficients directly from genome sequencing data. Methods to identify positively-selected (driver) mutations rely on finding genes that have significantly more mutational 'hits' (typically hits are non-synonymous mutations) than would be expected by chance, after correction for factors known to influence the mutation rate across the genome (*Bailey et al., 2018*). Conversely, negatively selected genes are expected to show a paucity of mutations (*Weghorn and Sunyaev, 2017*; *Zapata et al., 2018*). This idea is formalised in the calculation of the dN/dS ratio – a method originally developed in molecular species evolution – that has recently been adapted to study somatic evolution (both cancer and normal tissue) (*Martincorena et al., 2017*; *Weghorn and Sunyaev, 2017*; *Zapata et al., 2018*; *Wu et al., 2016*; *Greenman et al., 2006*; *Yang et al., 2003*; *Martincorena et al., 2018*; *Lee-Six et al., 2018*). The intuitive idea behind dN/dS is to measure the rate of non-synonymous (dN) mutations (possibly under selection) and compare that to the rate of synonymous (dS) mutations (presumed neutral). The ratio of these two numbers, each normalised for the local sequence-specific biases in the mutation rate, putatively identifies a signature of selection: dN/dS >1 indicating positive selection, dN/dS = 1 indicating neutral evolution and dN/dS <1 indicating negative selection.

Transforming dN/dS values to selective coefficients in somatic evolution is an unaddressed problem. dN/dS was originally developed in the context of species evolution using the Wright-Fisher process, a classical population genetics model that assumes that evolution occurs over very long timescales, which permits new mutations to fix within lineages. The Wright-Fisher model assumes constant population sizes, non-overlapping generations and that all individuals have equal potency. Under the Wright-Fisher model, the dN/dS of a locus is related to its selective coefficient by the relation (*Nielsen and Yang, 2003*):

$$\frac{dN}{dS} = \frac{2Ns}{1 - e^{-2Ns}}$$

Where $N$ is the effective population size and $s$ the selection coefficient.

However, in somatic evolution the assumptions of the Fisher-Wright model are violated. Somatic evolution is rapid and new mutations are infrequently fixed in the population (*McGranahan and Swanton, 2017*), clonal dynamics are complex (*Williams et al., 2019*), and population sizes unlikely to be constant (*Sottoriva et al., 2015*). Further, the lack of recombination in somatic evolution can result in strong hitchhiking effects (*Tilk et al., 2019*). In addition, since in somatic evolution, the ancestral genome is known, the need to measure dN/dS across a phylogeny is circumvented (a necessary step for dN/dS analysis in species evolution). Violations of some of these assumptions was previously recognised to make the interpretation of dN/dS problematic (*Kryazhimskiy and Plotkin, 2008*; *Mugal et al., 2014*), and consequently the relationship between selective coefficients and dN/dS values is uncertain.

The size distribution of clones (called the site frequency spectrum in population genetics nomenclature) also contains information on the selective coefficients of newly arising mutations. Mathematical descriptions of the dynamics of populations of cells can make predictions on the shape of the clone size distribution under different demographic and evolutionary models (*Simons, 2016a*; *Durrett, 2013*). This approach has been used to quantify the dynamics and cell fate properties of stem cells across many tissues in model systems (*Klein et al., 2010*; *Lopez-Garcia et al., 2010*; *Vermeulen et al., 2013*). We and others have also used similar approaches applied to deep sequencing data to infer the evolutionary dynamics of tumours (*Williams et al., 2016*; *Williams et al., 2018*; *Bozic et al., 2016*; *Ling et al., 2015*) and of clonal haematopoiesis in the blood (*Watson et al., 2019*).

To date, dN/dS analysis and the analysis of the clone size distribution have been performed independently, with conflictual results (*Simons, 2016b*; *Martincorena et al., 2016*). Here we develop the mathematical population genetics theory necessary to combine these approaches and explore how the inter-individual measure of selection at a locus as provided by dN/dS values, is related to the underlying cell population dynamics that generate intra-individual clone size distributions. This approach naturally accounts for the nuances in somatic evolution that can make the interpretation of dN/dS difficult. We show how this unified approach allows for greater insight into patterns of selection than either method in isolation, and importantly reveal the precise mathematical relationship

between dN/dS values and selective coefficients in somatic evolution. We use this approach to infer the selective advantage of mutations in normal tissue.

## Results

### A general approach to integrate dN/dS and clone size distributions

We present a general mathematical framework for the interpretation of frequency-dependent dN/dS values in somatic evolution. First, we construct null models of the evolutionary dynamics in the absence of selection, and then augment these models to incorporate the consequences of selection. Evolutionary dynamics differ between normal tissues and cancer cells: in normal tissues maintained by stem cells, the long-term population dynamics is controlled by an approximately fixed-size set of equipotent stem cells undergoing a process of neutral competition (*Klein and Simons, 2011*), whereas in tumour growth the overall population increases over time. We develop a null model to predict the expected genetic diversity in the population in the absence of selection. Positive selection causes selected variants to rise to higher frequency than expected under neutral evolution (*Figure 1a*), and negative selection has the opposite effect. This insight guides how we model the effects of selection, namely the diversity of non-synonymous mutations.

Specifically, we defined the function $g(\theta, \mu, s, f)$ as the expected number of mutations with selective (dis)advantage $s$ found at a frequency $f$, for a given evolutionary dynamics scenario, where mutations accumulate at a rate μ per division. For the remainder of the paper we use passenger mutations to refer to those mutations that have no functional effect (s=0) and driver mutations those that have s>0. When comparing to data, driver mutations are taken as equivalent to non-synonymous mutations and passengers equivalent to synonymous mutations.

The functional form of $g(\theta, \mu, s, f)$ encapsulates the population dynamics of the system with parameter vector $\theta$, which may, for example, include the growth rate of a tumour, or loss replacement rate of stem cells in normal tissue. The direct interpretation of $s$ depends on the system under question. Following the logic of the effect of selection above, for $s' > s$ we have that:

$$g\left(\theta, \mu, s', f\right) > g(\theta, \mu, s, f)$$

Since dN/dS measures the excess or deficiency of mutations due to selection, taking the ratio of $g(\theta, \mu, s, f)$ when $s \neq 0$ to $s = 0$ and normalizing for the mutation rates, which may differ for passenger ($\mu_p$) and driver ($\mu_d$) mutations respectively, informs how dN/dS is expected to change as a function of the frequency $f$ of mutations in the population (*Equation 1*).

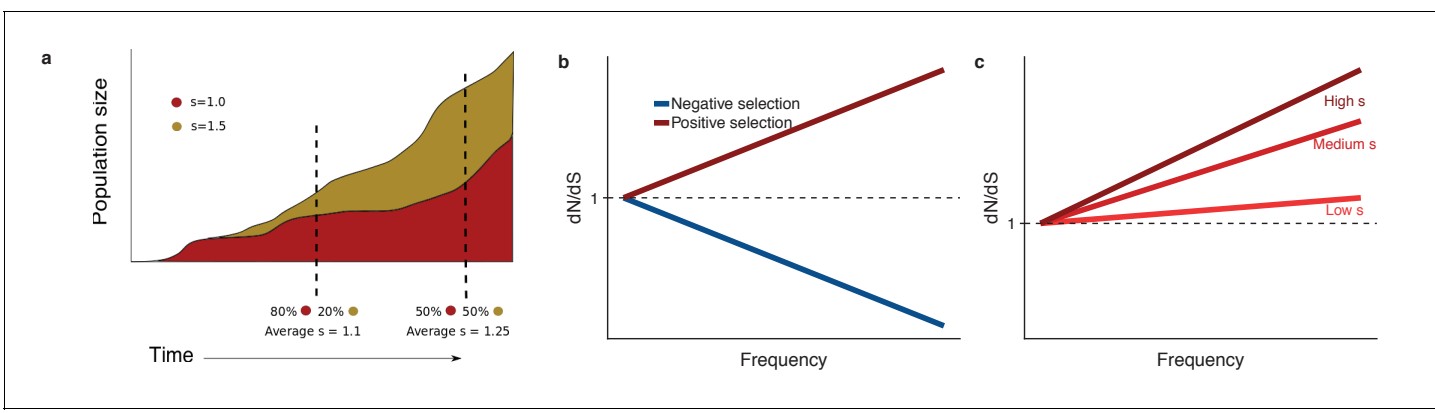

**Figure 1.** dN/dS in somatic evolution depends on the frequency of clones. (a) Variants under positive selection are enriched at high frequency, this means dN/dS estimates are dependent on the frequency of mutation, (b). The strength of selection influences the degree to which positively selected variants are enriched at high frequencies (c).

The online version of this article includes the following figure supplement(s) for figure 1:

**Figure supplement 1.** Global dN/dS values in different frequency bins for patient PD31182 showing that the values depend on the frequency of mutations.

$$\frac{dN}{dS} = \frac{\mu_p}{\mu_d} \frac{g(\theta, \mu_d, s, f)}{g(\theta, \mu_p, s=0, f)} \qquad (1)$$

We discuss the general properties of this model. Firstly, when $s = 0$ (neutral evolution), and provided that the mutation rates are correctly normalised, the numerator and denominator are equal resulting in $\frac{dN}{dS} = 1$, as expected. Secondly, dN/dS increases as a function of frequency $f$ (clone size) for positive selection, and decreases as a function of $f$ for negative selection (*Figure 1b*), for all $g(\theta, \mu, s, f)$ that we explored. Thirdly, the shape of the curves predicted by the underlying population model encodes the value of the selection coefficient; for example the steepness of the increase is proportional to the selection coefficient $s$ (*Figure 1c*). These observations are a natural consequence of positive selection driving selected mutations to higher frequency (*Figure 1a*).

Unfortunately, directly using *Equation (1)* to measure selective coefficients from the slope of the dN/dS curve as function of frequency is often impractical. Real sequencing data often suffers from a limited number of mutations detected at any particular frequency and measurement uncertainties in these frequencies. To circumvent these issues, we introduce 'interval dN/dS' (i-dN/dS) that aggregates over a frequency range to reduce the influence of these sources of noise. Interval dN/dS is defined as:

$$i\frac{dN}{dS} = \frac{\mu_p}{\mu_d} \frac{\int_{f_{min}}^{f_{max}} g(\theta, \mu_d, s, f) df}{\int_{f_{min}}^{f_{max}} g(\theta, \mu_p, s=0, f) df} \qquad (2)$$

Fixing the integration range $[f_{min}, f_{max}]$ allows for robust inference of $s$ in potentially sparse and noisy sequencing data using maximum likelihood methods (see Materials and methods).

## Frequency-dependent dN/dS values in stem cell populations

In healthy tissue, only mutations that are acquired in the stem cells will persist over long times, and so we restrict our attention to these cells. Quantitative analysis of lineage tracing data has shown that the stem cell dynamics of many tissues conform to a process of population asymmetry (*Klein and Simons, 2011*). In this paradigm, under homeostasis, the loss of stem cells through differentiation is compensated by the replication of a neighbouring stem cell, thus maintaining an approximately constant number of stem cells. These dynamics are represented by the rate equations:

$$SC \xrightarrow{2r\lambda} \begin{array}{l} SC + SC \\ D + D \end{array} \begin{cases} p = (1+\Delta)/2 \\ p = (1-\Delta)/2 \end{cases} \qquad (3)$$

where *SC* refers to a single stem cell which divides symmetrically to produce either two stem cells or two differentiated cells (denoted as *D* above), $\lambda$ is the rate of cell division per unit time, and $r$ is the probability of a symmetric divisions. The product $r\lambda$ is referred to as the loss/replacement rate. Differentiated cells will ultimately be lost from the population over long time scales. Under homeostasis, loss and replacement should be exactly balanced, so $\Delta = 0$. With $\Delta \neq 0$, the fate of a stem cell is 'biased', introducing positive or negative selection into the model. Previous mathematical analysis shows that this model is a good description of the clonal dynamics in the oesophagus and skin (*Klein et al., 2010*; *Doupé et al., 2012*; *Alcolea et al., 2014*). Using previous analytical results describing the temporal evolution of the clone size distribution (see methods for detailed discussion) we derive the frequency distribution $g(\theta, \mu, s, A)$ for oesophagus and skin as (*Simons, 2016a*; *Klein et al., 2010*; *Nicholson and Antal, 2016*):

$$g(\theta, \mu, s, A) = \frac{\mu n_0}{r\lambda \rho A} e^{-\frac{\rho A}{N(t)}} \qquad (4)$$

Where A is the area of the clone, $\rho$ is density of stem cells per mm$^2$, $n_0$ is the starting population size and μ the mutation rate, which may be different for drivers ($s > 0$) and passenger mutations ($s = 0$), ie drivers and passengers may accumulate at different rates. $N(t)$ is a scaling factor that depends on $\Delta$, the bias toward self-renewal, which we interpret as our selection coefficient in this system. Specifically:

$$N_{\Delta=0}(t) = 1 + r\lambda t \qquad (5)$$

$$N_\Delta(t) = \frac{(1+\Delta)e^{2r\lambda\Delta t} - (1-\Delta)}{2\Delta} \qquad (6)$$

$N(t)$ can be interpreted as the average size of a labelled clone after time $t$, which even under homeostasis grows over time and compensates for some clones being lost due to drift. From these expressions, we can then write down an expression for i-dN/dS as a function of clone frequency (see Materials and methods) that allows for maximum likelihood estimation of parameter values ($\Delta$). We confirmed the accuracy of our derivation using simulations (*Figure 2a*), and performed power calculations to determine the minimum number of mutations required to correctly infer the underlying population dynamics: we determined that 8 mutations per gene was sufficient to accurately recover $\Delta$ (*Figure 2b*) with accuracy increasing for higher mutation burdens (*Figure 2c*). We also performed simulations where $\Delta$ was itself a random variable, simulating the effect of different sites within the gene having different fitness effects. We assumed $\Delta$ was exponentially distributed and generated 500 simulated cohorts. Fitting i-dN/dS demonstrated that on average we infer the mean value of the exponential distribution, *Figure 2—figure supplement 1*.

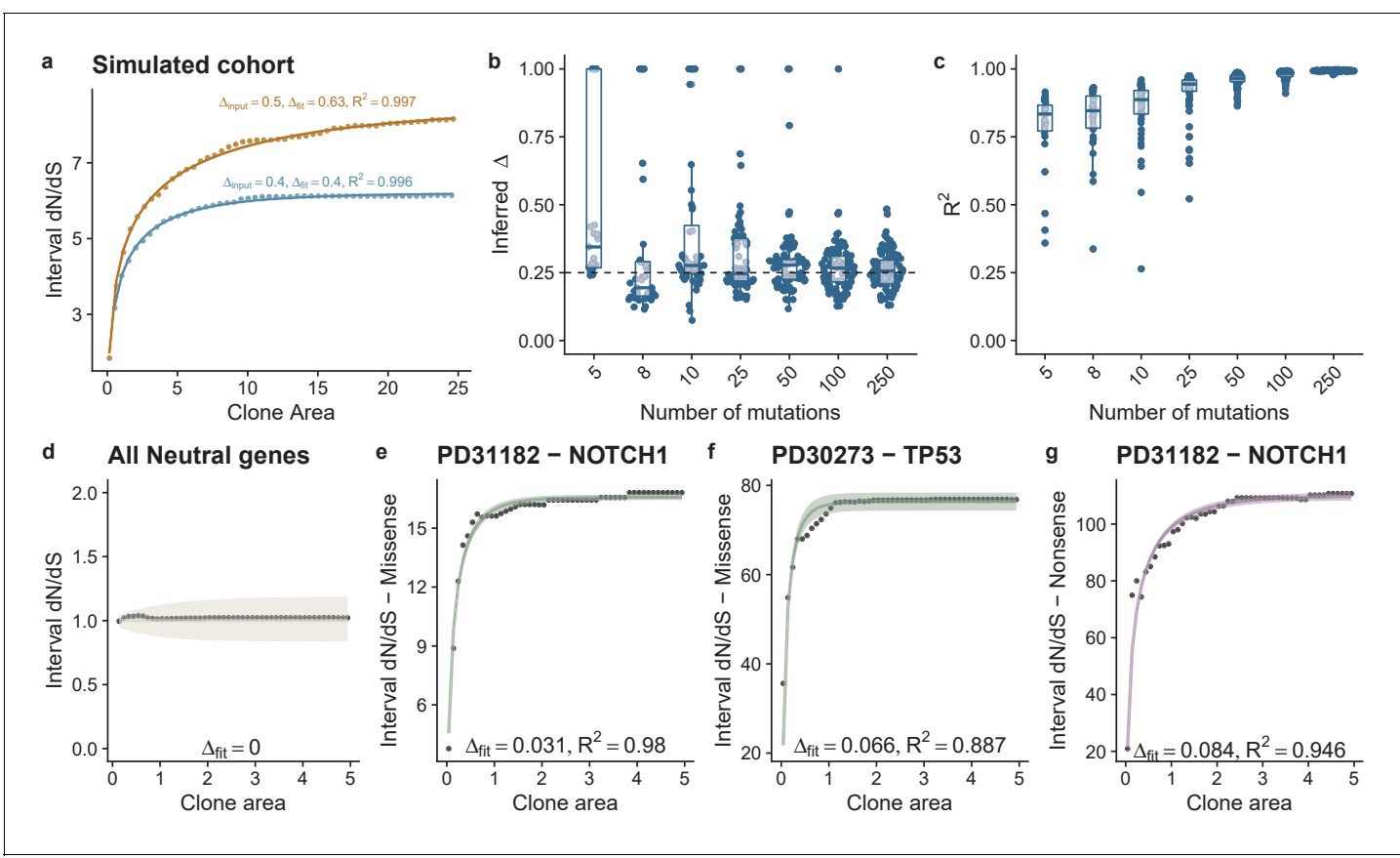

**Figure 2.** Theoretical model of interval dN-dS fitted to simulated data and data from deep sequencing of the oesophagus. (a) Interval dN/dS as a function of clone area for 2 simulated cohorts where driver mutations induce different biases, theoretical model captures the dynamics well and enables us to recover the bias $\Delta$, accurately. As the number of mutations increases ability to recover the correct $\Delta$ and the model fit (measured using $R^2$) improves (b) and (c). (d) Data and model fit for all neutral genes, shows i-dN/dS = 1 across the frequency range and inferred bias of 0. Data and model fit for (e) NOTCH1 missense mutations in patient PD31182, (f) missense TP53 mutations in PD30273 and NOTCH1 nonsense mutations in PD31182 (g). Data are black points and model fits are solid lines with shaded areas denoting 95% CI.

The online version of this article includes the following figure supplement(s) for figure 2:

**Figure supplement 1.** Histogram of inferred $\Delta$ values from simulations using an exponentially distributed fitness effect.
**Figure supplement 2.** Model fits for all patients in the oesophagus data set.
**Figure supplement 3.** Inferred biases for for each patient in the oesophagus dataset based on missense, (a) and nonsense mutations, (b).
**Figure supplement 4.** Individual fits for each gene in each patient in the oesophagus dataset.

## Selection advantages in histopathologically-normal human oesophagus

We inferred the selective advantage of driver mutations in human oesophagus using published deep sequencing data from Martincorena and colleagues (*Martincorena et al., 2018*) that documents the clonal expansion of a panel of putative driver mutations in histopathologically-normal oesophageal biopsies.

We used the dndscv bioinformatics tool (*Martincorena et al., 2017*) to calculate frequency-dependent dN/dS values from these data (clone size measured in fraction of mutant reads multiplied by 2 mm$^2$ – the area of the biopsy – and assuming 5000 stem cells per mm$^2$ (*Eyre-Walker and Keightley, 2007*) tissue). dN/dS values varied considerably as a function of mutation area (*Figure 1—figure supplement 1*).

We considered the average frequency-dependent dN/dS values simultaneously across all genes in the panel, on a patient-by-patient basis. Our theoretical model of i-dN/dS calculated from these data fitted strikingly well (*Figure 2—figure supplement 2*). Estimates of the loss/replacement rate $r\lambda$ of the stem cell population were in the range 1.2-5.0 per year (*Figure 2—figure supplements 2* and *3*). Inference of the selective advantage $s$ (measured in terms of the bias towards self renewal $\Delta$) revealed an average bias of 0.004 (0.002 – 0.005 95% CI) per missense mutation (*Figure 2—figure supplement 3*). Nonsense mutations caused a five-fold greater bias towards self-renewal of 0.021 (0.008 – 0.032 95% CI) (*Figure 2—figure supplement 3*). After removal of all genes that are strongly selected, global dN/dS values on the remaining 48 genes show dN/dS of approximately 1 across the frequency range (*Figure 2d*), and i-dN/dS analysis revealed that these somatic mutations do not associate with a proliferative bias ($\Delta$=0).

We then fitted the data on a gene-by-gene and patient-by-patient basis for cases where sufficient mutations were available to perform the fit (*Figure 2e–g*; *Figure 2—figure supplement 4*). A broad range of selective advantages were inferred (*Figure 3* and *Figure 3—figure supplement 1*). Mutations in *TP53* showed large biases across all patients for both missense, $\Delta$ = 0.057 (0.05–0.068 95% CI) and nonsense mutations, $\Delta$ = 0.094 (0.091–0.097 95% CI) (*Figure 3a–b*). This was also true for mutations in NOTCH1 with $\Delta$ = 0.029 (0.019–0.036 95% CI) for missense and $\Delta$ = 0.072 (0.034–0.089 95% CI) for nonsense mutations. *NOTCH2*, *PIK3CA*, *CREBBP* and *FAT1* also showed a bias toward self-proliferation in multiple patients (*Figure 3a–b*), though most had a small effect on fitness (range 0.003–0.029 for missense mutations and 0.030–0.041 for nonsense mutations). Together these data suggest a distribution of fitness effects (DFE) characterized by many small effect mutations with few large effect mutations (*Figure 3c–d*), as is seen in organismal evolution (*Eyre-Walker and Keightley, 2007*). We recognize that there may be intra-gene variation of selection coefficients, that is, some sites within genes may have stronger fitness effects than others. This is supported by clustering of mutations within particular domains and hotspots of mutations as documented in the original study (*Martincorena et al., 2018*). In future, larger cohorts and methods to estimate site level dN/dS values would allow this approach to be extended to the site level.

As our model assumes that clones emerge and expand independently we checked that the data is not overly influenced by hitchhiking mutations,which would violate these assumptions. For this, we leveraged the spatial sampling of tissue pieces. Approximately 90 patches were sampled from each patient. We reasoned that patches with selected clones might be expected to have more hitchhiking mutations, and for those mutations to be at a higher frequency when compared to patches without selected clones. To test this hypothesis, we counted the number of non-synonymous NOTCH1 and TP53 mutations and the number of synonymous mutations in each patch. If the synonymous mutations we observe in the data are largely due to hitchhiking effects we would expect the number and size of synonymous variants to correlate with the number of driver mutations per patch. While there =was a statistically significant correlation for both NOTCH1 (linear regression, p<0.001) and TP53 mutations (p=0.031), the effect was small (*Figure 5—figure supplement 1*): for each additional driver there was on average 0.05 additional synonymous variants (0.047 for NOTCH1 and 0.056 for TP53). We note too that the correlation was very noisy ($R^2 < 0.02$) and we observed no statistically significant relationship between VAF of synonymous mutations and the number of TP53 or NOTCH1 mutations (linear regression, p>0.1). This analysis suggests that the majority of synonymous mutations are not hitchhikers, and consequently that assuming the independence of clones isreasonable.

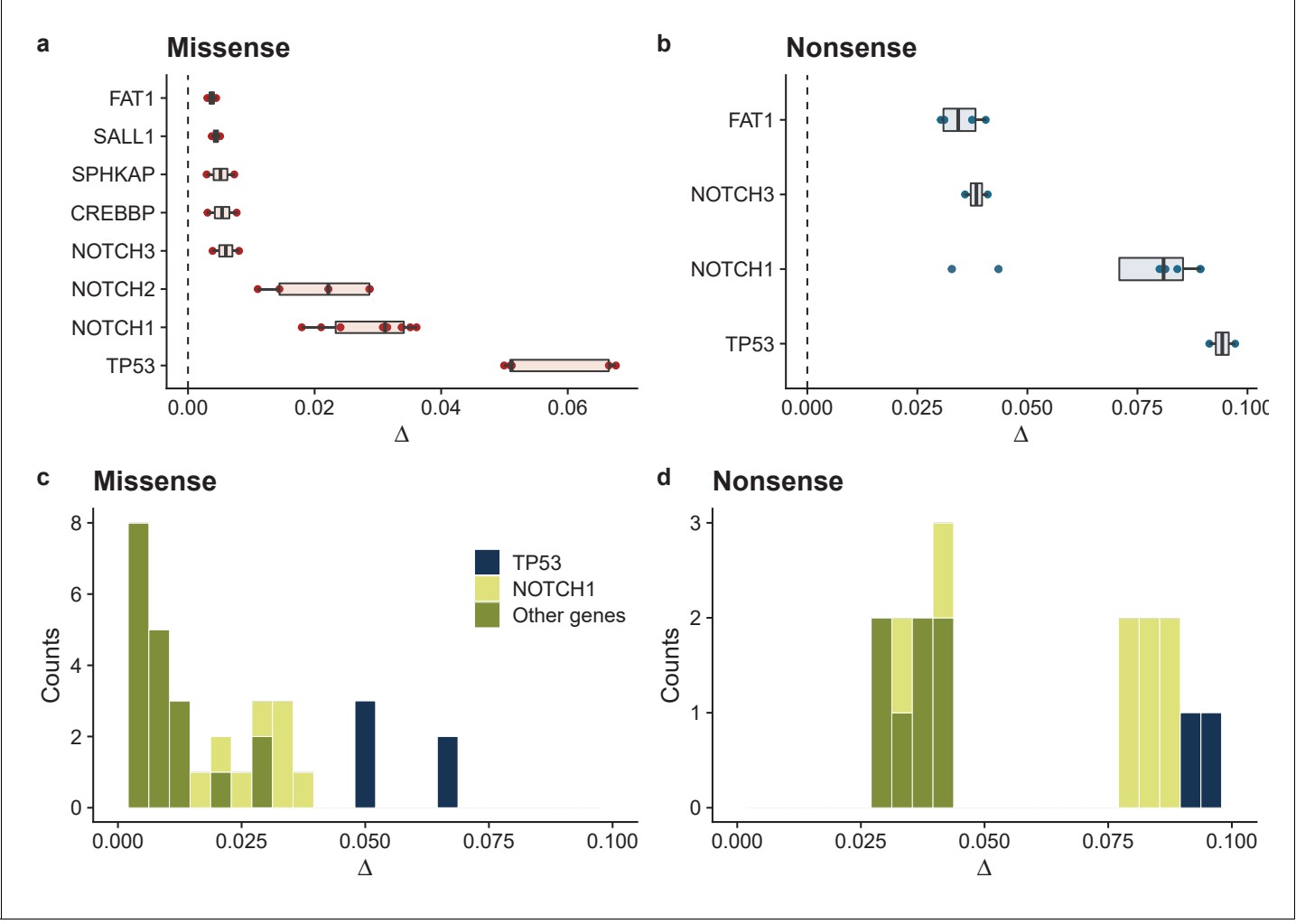

**Figure 3.** Summary of model fits across all patients for normal oesophagus data. Inferred biases Δ for genes where at least 2 patients had good model fits ($R^2 > 0.6$ & >7 mutations) for missense mutations (**a**), and nonsense mutations (**b**). Inferred distribution of fitness effects for all genes across all patients for missense mutations (**c**), and nonsense mutations (**d**).

The online version of this article includes the following figure supplement(s) for figure 3:

**Figure supplement 1.** Inferred parameters for each gene in each patient in the oesophagus dataset where there were sufficient mutations to perform the analysis.

## Driver mutation selective advantage in normal skin

Martincorena and colleagues had also published data on the expansion of driver mutations in ostensibly normal human skin (*Martincorena et al., 2015*). Analyses of these data with interval dN/dS revealed a per-patient average selective advantage per mutation (again measured in terms of the bias towards self renewal) of Δ = 0.001 for missense mutations and four-fold higher (Δ = 0.004) for nonsense mutations (*Figure 4a-c*). Performing the analysis on a gene-by-gene basis was limited by the lower number of detected mutations, and the limited frequency range (clone size range) compared to the oesophagus dataset. Good fits to the data were obtainable for *NOTCH1* missense mutations in patient PD18003 with fitness estimated to be Δ = 0.0149 (0.0148-0.0150 95% CI), and *TP53* missense mutations also in patient PD18003, Δ = 0.0054 (0.0051-0.0058 95% CI) *Figure 4*. These fitness coefficients were similar to the oesophagus data. For missense mutations we were also able to produce the distribution of fitness effects across the skin cohort, which showed similar

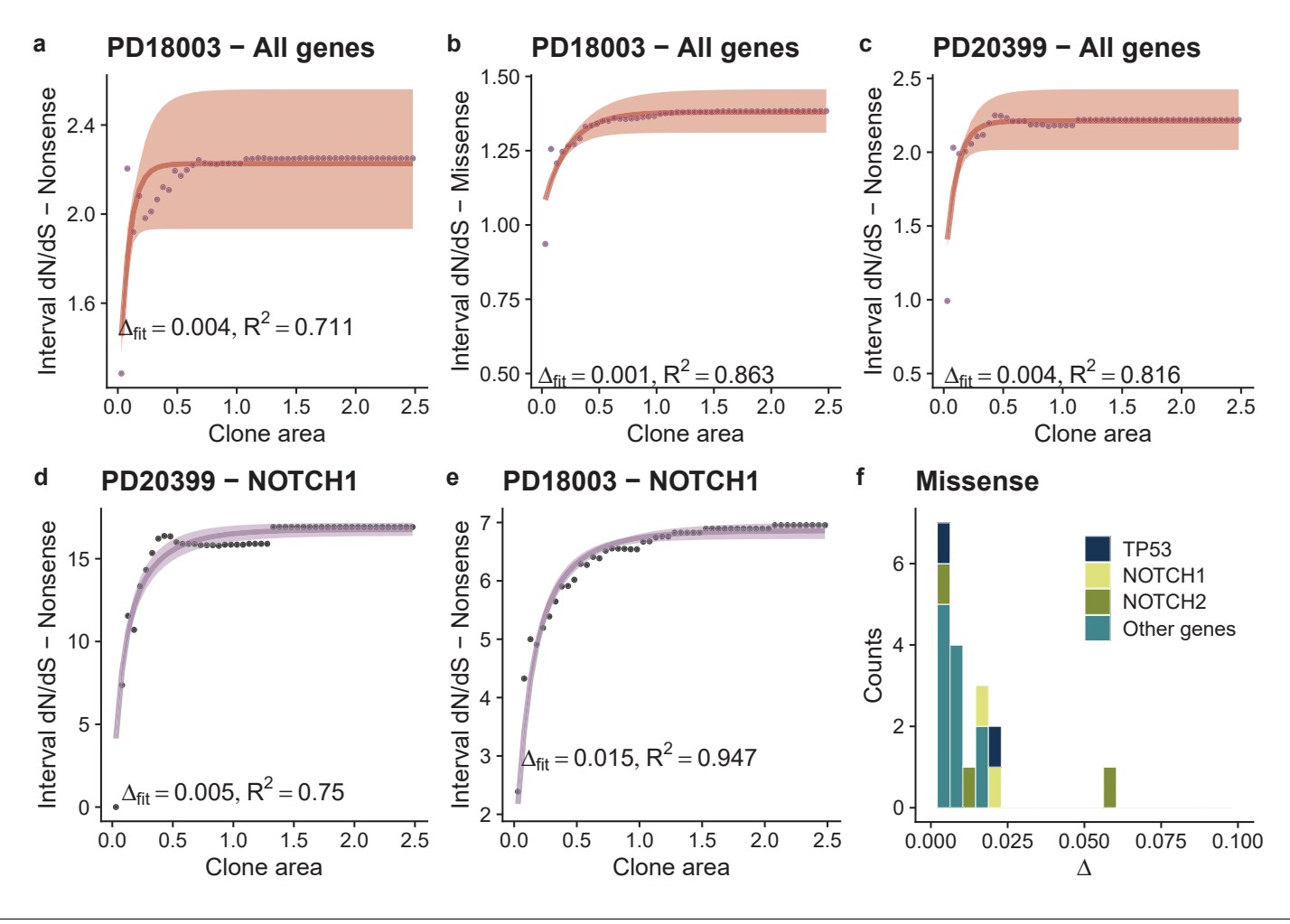

**Figure 4.** Analysis of skin dataset shows similar DFE to oesophagus. Model fits per patient and per gene per patient when there were sufficient mutations in the skin dataset. Points are data and lines are model fits, (a-e). (f) Shows the distributions of fitness effects for missense mutations across the cohort. There were insufficient nonsense mutations in the majority of genes to draw the equivalent plot for nonsense mutations.

characteristics to the oesophagus data of a small number of high effect mutations and a larger number of smaller effect mutations, *Figure 4f*.

## Site frequency spectra

We next sought to challenge our model by directly fitting the site frequency spectra across ages, taking a similar approach to studies of the blood (*Watson et al., 2019*), colon (*Lopez-Garcia et al., 2010*), skin (*Simons, 2016a*) and other tissues (*Klein and Simons, 2011*). Our model of stem cell dynamics makes predictions on the properties of these distributions as a function of the age of donors. In particular, *Equations (5) and (6)* predict that the characteristic frequency N(t) increases exponentially for non-neutral mutations and linearly for neutral mutations as a function of time. Plotting the distribution of clone size areas showed a widening of the distribution as a function of age, which was particularly striking for mutations in the NOTCH1 and TP53 genes, consistent with these genes conferring large selective advantages (*Figure 5a*).

To quantitively test the predictions of the model and to infer parameters of interest we implemented a Bayesian non-linear fitting method (see methods) to fit the following model:

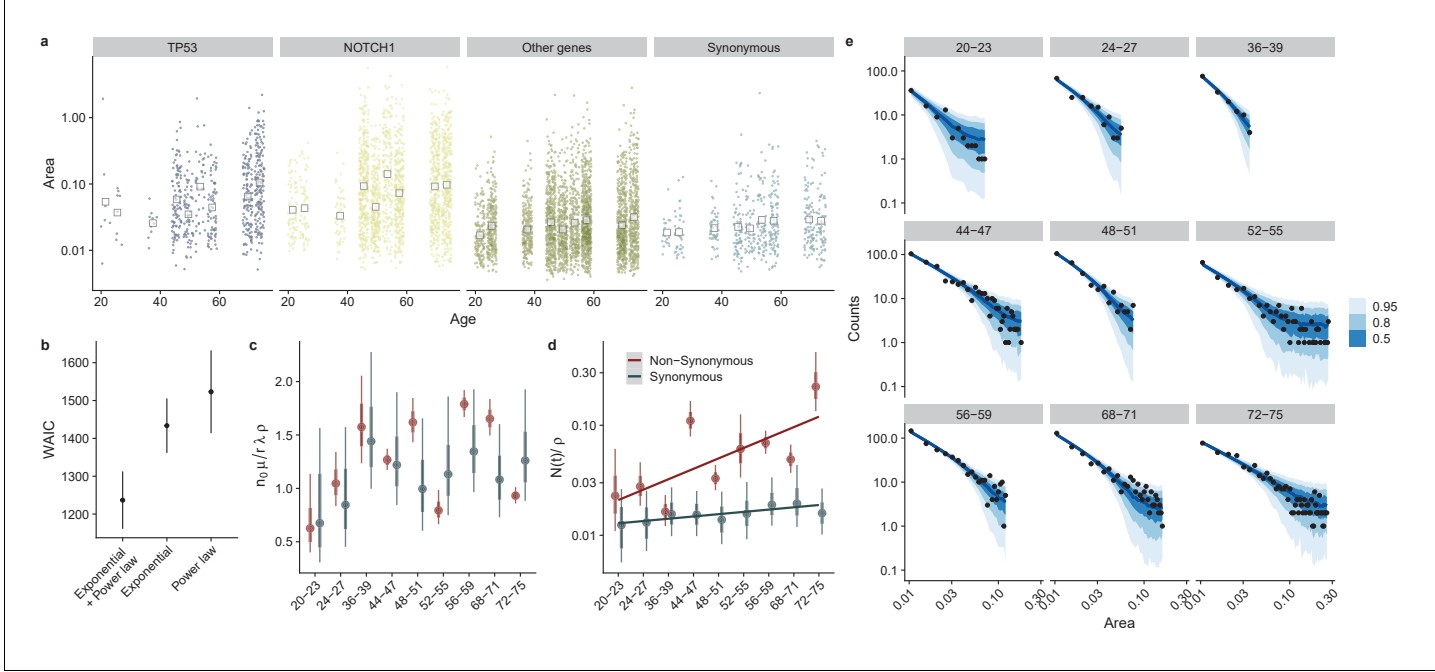

**Figure 5.** Directly fitting site frequency specta supports interval dN/dS inferences. (**a**) Site frequency spectra become wider for older donors, with increases in the median clone area which is more pronounced for mutations in TP53 and NOTCH1. (**b**) Using WAIC to perform model selection, we found a model with an exponential term and a power law to be the best fitting model (lowest WAIC). (**c**) Posterior parameter estimates for $n_0\mu$. (**d**). The characteristic frequency $N(t)/\rho$, Interval represent 66% and 95% credible intervals respectively. (**e**) Site frequency spectra from data (black dots) and posterior predictive fits for 50, 80% and 95% credible intervals (blue ribbons) for non-synonymous mutations in each don or.

The online version of this article includes the following figure supplement(s) for figure 5:

**Figure supplement 1.** The number of synonymous mutations as a function of the number of NOTCH1 (a) and TP53 (b) mutations per tissue piece.

**Figure supplement 2.** Inferring parameters from the site frequency spectrum of a simulated dataset.

**Figure supplement 3.** Site frequency spectra fits for NOTCH1 and TP53 non-synonymous mutations, (a).

**Figure supplement 4.** Regression of clone size against age.

**Figure supplement 5.** Comparison of results using dndscv and SSB-dN/dS.

$$y = \frac{\alpha}{A} e^{-\frac{A}{\exp(\beta)}} \tag{7}$$

With the parameters $\alpha = \mu n_0/r\lambda\rho$, and $\beta = \log(N(t)/\rho)$ to be estimated. We first validated that the approach could correctly infer known parameters from synthetic data generated from our simulation framework, *Figure 5—figure supplement 2*. Next, we fitted the model to the oesophagus dataset, separately for non-synonymous and synonymous mutations and across ages, *Figure 5e*. Posterior estimates of the compound parameter $\mu n_0/r\lambda\rho$ showed consistent estimates for both non-synonymous and synonymous mutations (taking into account the approximate 3:1 ratio of mutable sites), *Figure 5c*. Unfortunately decoupling $\mu n_0/r\lambda\rho$ directly from the data is not possible and requires independent estimates of either the mutation rate or number of stem cells. Posterior estimates of the characteristic frequency N(t) showed an increase as a function of age for non-synonymous mutations and a more modest increase for synonymous as would be predicted from theory, *Figure 5d*.

As another challenge to the proposed model, we also fitted the following two models which are simpler subsets of the full model we derive theoretically, thus testing two alternate decay functions.

$$y = \frac{\alpha}{A}$$

$$y = \alpha e^{-\frac{A}{\beta}}$$

Here, $\alpha$ and $\beta$ are parameters to be estimated. Comparison of the fits using the widely applicable information criteria, WAIC (a generalized version of information criteria such as AIC and BIC) found that our theoretical model with an exponential term and a power law term (*Equation 7*) provided the best predictive accuracy (lowest WAIC), Figure 5b. Bayes factors for the proposed theoretical model also strongly supported this as the best model (BF > $10^3$ for both models).

We also attempted to fit the model on a gene by gene basis. However due to the limited data points at the gene level we found that our posterior estimates for the parameters were very wide, precluding further insight from the site frequency spectrum at the gene level, *Figure 5—figure supplement 3*. This reveals one of the strengths of our i-dN/dS approach, which by leveraging information across genes to infer the background mutation model and integrating over clone sizes we can perform the inference with a limited number of data points. As an alternative to fitting the full clone size distribution, we performed a Bayesian multi-level regression to assess which mutations showed the largest increase in clone size as a function of age. This analysis revealed that TP53 and NOTCH1 had the largest regression coefficients, consistent with these genes having the largest selection coefficients, *Figure 5—figure supplement 4*, providing qualitative support for our approach. Our inferred selection coefficients from i-dN/dS were also correlated with the regression coefficients from the statistical model (linear regression, p=0.004, $R^2$ = 0.47). Taken together, analysis of the site frequency spectrum on a patient by patient basis and a gene level statistical model are consistent with our inferences from i-dN/dS. We find that the proposed model provides the best fit to data when compared to other similar models, that the characteristic frequency of non-synonymous mutations increases rapidly with age and that NOTCH1 and TP53 exhibit the largest increase in clone size as a function of age.

## Discussion

Here we have shown that the combination of dN/dS values with mutation frequency-based information provides additional quantitative insight into dynamics of somatic evolution than either method alone. Specifically, the combined approach enables direct inference of the selection coefficients of mutations in somatic tissues. We note that our study also shows that the magnitude of the selection coefficient is not necessarily represented by simply calculating a 'point estimate' of dN/dS that neglects mutation subclonal frequency information.

Using this methodology, we have begun the construction of the distribution of fitness effects (DFE) in somatic evolution, observing a distribution where most mutations that we analysed were near-neutral, with a tail of highly selected variants. In both skin and oesophagus, the most highly selected mutant genes were *NOTCH1* and *TP53* (increased proliferative bias of >1% and>5% respectively). We observed that values of selective coefficients of individual genes varies between patients, likely because of inter-patient difference in the precise location of point mutations, but potentially also because of inter-patient variation in selective pressure from the microenvironment. Nevertheless, the comparative rank of per-gene fitness coefficients was broadly consistent across patients (e.g. for missense mutations across patients, *TP53* mutations always had the highest fitness, followed by *NOTCH1* mutations). This consistency in selective coefficients is in agreement with the observation of highly recurrent gene mutations in cancer (*Lawrence et al., 2013*) and evidence of repeatability in cancer evolution (*Caravagna et al., 2018*).

Non-synonymous *NOTCH1* mutations were observed approximately 3-fold more frequently that non-synonymous *TP53* mutations in oesophagus, and approximately 5-fold more frequently in skin, suggesting that the mutation rate of *NOTCH1* is greater than for *TP53*. Coupling these data with our quantitative measurements of the fitness coefficients leads to the prediction that the oesophagus will become transiently repopulated by *NOTCH1*-mutant cells during ageing, before subsequent replacement by fitter mutant-*TP53* clones.

On a cautionary note, our theoretical work shows that the clonality of mutations strongly determine the observed value of dN/dS, and so a misleading picture of the selective forces will be produced if dN/dS frequency-dependent effects are not corrected for. The accuracy of any estimate of evolutionary dynamics from dN/dS values is of course dependent on the underlying accuracy of the dN/dS measure itself, which is compromised by uncharacterised variability in the mutation rate across the genome (*Van den Eynden and Larsson, 2017*) and in the uncertain pathogenicity of individual single nucleotide variants (extensions to estimate site level selection coefficients may

circumvent some of these issues [*Cannataro et al., 2018*; *Temko et al., 2018*]). We also recognize that our model assumes a well-mixed population, while the data used in our study is from spatially structured epithelia. Spatial structure influences the distribution of clone sizes. Effects include: the influence of the boundary that enable rapid growth of mutants on the expanding front and conversely 'encapsulation' within a growing mass that slows clone growth (*Fusco et al., 2016*; *Chkhaidze et al., 2019*), and clonal interference that slows the growth of two similarly fit competing clones (*Martens et al., 2011*; *Hall et al., 2019*).

Combining population genetics methods with comparative genomics is a powerful way to infer selection pressures in human somatic evolution, giving new insight into the fundamental parameters that determine evolutionary dynamics in health and disease.

## Materials and methods

### Oesophagus and skin data

For the oesophagus and skin data we used mutation calls provided by the original studies. In the oesophagus data when a mutation was present in multiple adjacent biopsies we used the sum of the mutation frequency times the area of the biopsies (2 mm$^2$) as our readout of clone size and performed the dN/dS analysis on a patient by patient basis.

### dN/dS calculations

For calculating dN/dS ratios the dndscv R package was used, which calculates both global dN/dS ratios across the whole exome or a panel of genes as well as per gene dN/dS ratios using a covariate based model to infer dN/dS values with a limited number of mutations (*Martincorena et al., 2017*). We used the default settings of dndscv, using the default hg19 transcript reference provided by the package. dndscv can also take into account small insertions and deletions, which we included in our analysis. Therefore, where we refer to mutation this includes indels in addition to SNVs.

To calculate the interval dN/dS measure we took the clone size measurements and determined a low cutoff $f_{min}$ based on the minimum clone size. We then created a vector of clone sizes that covered the total range and calculated dN/dS between $f_{min}$ and all values of $f_{max}$. This allowed us to plot dN/dS as a function of $f_{max}$ and fit our interval dN/dS models. In our data, clone size is measured in units of area (mm$^2$).

Accurately estimating dN/dS from sequencing data of somatic tissues can be challenging due to the strong sequence context dependence of mutations and variability of mutation rates across the genome. To confirm our inferences were not dependent on the choice of dN/dS methodology we calculated dN/dS values using SSB-dNdS and then fitted our model (*Zapata et al., 2018*). As SSB-dNdS only uses SNVs, we reran dndscv after removing indels. Inferences on a patient by patient basis were highly consistent between the methods, see *Figure 5—figure supplement 4*. There was more variability at the gene level, perhaps due to differences in the approaches used to control for variability in mutation rates across the genome.

### Model fitting

We used a maximum likelihood approach to fit our models to the data. Defining the observed interval dN/dS as $y$ and the model dN/dS as $\hat{y}(\theta) = \frac{\mu_p}{\mu_d} \frac{\int_{f_{min}}^{f_{max}} g(\theta, \mu_d, s, f) df}{\int_{f_{min}}^{f_{max}} g(\theta, \mu_p, s=0, f) df}$. First of all we define the residuals between the data and the model as $R = y - \hat{y}(\theta)$. Assuming that the residuals are normally distributed with mean 0 we can write down the negative log likelihood (NLL) as

$$NLL(\theta) = -\sum_{y - \hat{y}(\theta)} \log N(y - \hat{y}(\theta), \mu = 0, \sigma)$$

where $N$ denotes the normal probability density function. We can then find the parameters $\theta$ that minimize the NLL and calculate confidence intervals on these estimates using the Fisher information matrix. When fitting to data we ensured that there were a minimum of 8 mutations and only included model fits with R$^2$ > 0.6 for downstream analysis. We used a maximum likelihood approach over a Bayesian approach to fit the model because the integral of the clone size distribution does not have

a closed form solution, making it unfeasible to use readily available MCMC samplers which we adopt later.

## Interval dN/dS model

For the stem cell model, using *Equations 2, 3, 4, 5, 6* in the main text, interval dN/dS is given by:

$$i\frac{dN}{dS} = \frac{1}{1+\Delta} \frac{\left[E_i\left(-\frac{\rho A_{max}}{N_\Delta(t)}\right) - E_i\left(-\frac{\rho A_{min}}{N_\Delta(t)}\right) + \frac{1}{2}\left(\frac{e^{\frac{\rho A_{max}}{N_\Delta(t)}}}{\rho A_{max}} + \frac{e^{-\frac{\rho A_{min}}{N_\Delta(t)}}}{\rho A_{min}}\right)\right]}{\left[E_i\left(-\frac{\rho A_{max}}{N(t)}\right) - E_i\left(-\frac{\rho A_{max}}{N(t)}\right) + \frac{1}{2}\left(\frac{e^{\frac{\rho A_{max}}{N(t)}}}{\rho A_{max}} + \frac{e^{-\frac{\rho A_{min}}{N(t)}}}{\rho A_{min}}\right)\right]}$$

Where $E_i$ is the exponential integral $E_i(x) = -\int_x^\infty \frac{e^{-n}}{n}dn$. $\rho$ is density of stem cells per mm$^2$, which we set to 5,000 cells /mm$^2$ for fitting (*Hall et al., 2019*).

## Simulations

To confirm the accuracy of our analytical model and investigate the influence of uncertainty in mutation frequencies due to sequencing noise and to challenge some of the underlying assumptions of our theoretical approach, we developed a simulation based model.

We seed a population of $N_s$ stem cells that then undergo loss/replacement as described by the following rate equations

$$SC \xrightarrow{2r\lambda} \begin{matrix} SC + SC \\ D + D \end{matrix} \quad \begin{cases} p = (1+\Delta)/2 \\ p = (1-\Delta)/2 \end{cases}$$

As only the stem cells are long lived the differentiated cells are not explicitly modelled such that when a stem cell 'differentiates' it is effectively lost from the population. During division, daughter cells acquire mutations with a fitness effect at rate $\mu_d$ and passenger mutations at rate $\mu_p$. Fitness increases the bias toward self-proliferation $\Delta$ of a stem cell lineage. Additional driver mutations do not further increase the fitness of stem cells.

To calculate dN/dS across a cohort of tissue biopsies we count the number of driver mutations $N_d$ and the number of passenger mutations, $N_p$ and then normalize by their respective mutation rates. In our model drivers = non-synonymous and thus every driver has an effect on fitness. Then the ratio of these two numbers gives us the excess or deficit of mutations due to selection – ie the dN/dS ratio.

$$\frac{dN}{dS} = \frac{N_d/\mu_d}{N_p/\mu_p}$$

For the interval dN/dS we simply calculate the $N_x$ between $f_{min}$ and $f_{max}$.

To introduce uncertainty into mutation frequencies we perform a process of empirically motivated sampling to the true underlying frequency $f$. Firstly, we specify the average depth of sequencing D, then the depth of sequencing for mutation i is given by

$$D_i = Poisson(D)$$

The sampled number of read counts is then

$$n_s = Binomial(n = D_i, p = f)$$

And the sampled variant frequency is then $f_s = n_s/D_i$.

The simulation framework was written in Julia (*Bezanson et al., 2017*) and is available at https://github.com/marcjwilliams1/StemCellModels.jl.

## Fitting site frequency spectra

To fit the site frequency spectra we first removed mutations with clone size area < 0.008. This cutoff was determined by inspection of the point of highest density of the clone size area histograms, reasoning that below this frequency the data was limited due to the resolution of the sequencing assay. We then binned the data using a bin size of 0.005 and counted the number of mutations in each bin. We did this for each donor and separately for non-synonymous and synonymous mutations. We then

used the brms (Bayesian Regression Modeling with Stan) R package (*Bürkner, 2017*) to fit the following non-linear model jointly across all patients:

$$C = \frac{\alpha}{A} e^{-\frac{A}{\exp(\beta)}}$$

With the parameters $\alpha = \frac{\mu n_0}{r \lambda \rho}$, and $\beta = \log(N(t))$ to be estimated and C being the number of mutations and A the area. We used the following priors for the parameters:

$$A \sim Normal(5, 2)$$

$$B \sim Normal(0, 5)$$

We ran 4 chains with 5000 iterations with the first 2500 used as warmup. To assess convergence we ensured that the scale reduction factor $\hat{R}$ (a measure of mixing of chains) was < 1.01 as recommended.

We also fitted the following two models:

$$C = \frac{\alpha}{A}$$

$$C = \alpha e^{-\frac{A}{\exp(\beta)}}$$

We used the same prior distributions as above. We then compared the predictive accuracy of the different models using the widely applicable information criteria (WAIC) (*Vehtari et al., 2017*). This found that the functional form derived from theory provided the best fit, *Figure 5b*. Bayes factors to compare models were calculated using the bayestestR package (*Makowski et al., 2019*). The log10 (BF) of the proposed theoretical model vs the power law was 52 and 44 when compared against the exponential model.

## Bayesian multi-level regression

We performed a Bayesian multi-level regression of clone size ~ age with the gene as a random effect. This allowed us to determine which genes cause the largest increase in frequency as a function of age. This was done using brms in R with default priors, 4 chains and 5000 iterations. Using R's statistical modelling syntax the model is given by *clone size ~age + (1 + age|gene)*. We fit the model assuming a normal distribution as well as a log-normal distribution, finding the latter to provide the best predictive accuracy (lowest WAIC) and superior posterior predictive check.

## Site frequency spectra for a model of stem cell proliferation

Our mathematical model of stem cell proliferation drew on results from a range of studies analysing the clone size distribution from lineage tracing experiments. Particularly useful are the results from *Klein et al. (2010)*, which we took as a starting point for our model. We follow the notation used in this study to a large extent. Other studies that are also illuminating and relevant are the theoretical work of *Nicholson and Antal (2016)*, while similar theoretical models have also been applied to the oesophagus (*Doupé et al., 2012*), airway epithelia (*Teixeira et al., 2013*) and the blood (*Watson et al., 2019*) amongst others. In this section we outline the key results relevant to our approach.

We begin with the set of rate equations presented in the main text:

$$SC \xrightarrow{2r\lambda} \begin{matrix} SC + SC \\ D + D \end{matrix} \quad \begin{cases} p = (1 + \Delta)/2 \\ p = (1 - \Delta)/2 \end{cases}$$

From this we are interested in the clone size distribution, that is, the probability of observing a clone of size n after time t. In other fields such as population genetics, this distribution is equivalent to the site frequency spectrum. Given the above rate equations, we can express the dynamics as a birth-death process with birth rate $b = r\lambda(1 + \Delta)$ and death rate $d = r\lambda(1 - \Delta)$ which allows us to write down the master equation for the probability of observing a clone of size n at time t, $p_n$.

$$\frac{dp_n}{dt} = b(n-1)p_{n-1} - (b+d)np_n + d(n+1)p_{n+1}$$

$$\frac{dp_0}{dt} = dp_1$$

This has the following solution (see *Bailey, 1990* p92)

$$p_n(t) = (1 - \alpha(t))(1 - \beta(t))\beta(t)^{n-1}$$

$$p_0(t) = \alpha(t)$$

With $\alpha(t)$ and $\beta(t)$ defined as follows:

$$a(t) = \frac{d\left(e^{(b-d)t} - 1\right)}{be^{(b-d)t} - d}$$

$$\beta(t) = \frac{b\left(e^{(b-d)t} - 1\right)}{be^{(b-d)t} - d}$$

From these results we can obtain the average size of surviving clones, $N(t)$:

$$N(t) = \sum_{n=0}^{\infty} n \times \frac{p_n}{1 - p_0} = \frac{1}{1 - \beta(t)} = \frac{(1+\Delta)e^{2r\lambda\Delta t} - (1-\Delta)}{2\Delta}$$

Which, as will become apparent, gives a characteristic scale for the distribution.

Thus far we have only considered a single mutant at time 0, while we are interested in the case when mutants continually enter the population at a rate $\mu n_0$ where μ is the mutation rate per cell division and $n_0$ is the number of stem cells. To derive the clone size distribution in this case, we take the integral over time multiplied by the mutation rate.

$$g_n = \frac{\mu n_0}{r\lambda} \int_0^t p_n(\tau)d\tau = \frac{\mu n_0}{r\lambda n} \beta(t)^n$$

We can approximate $\beta(t)^n = (1 - 1/N(t))^n \approx e^{-\frac{n}{N(t)}}$ which gives

$$g_n \approx \frac{\mu n_0}{r\lambda n} e^{-\frac{n}{N(t)}}$$

The data we make use of doesn't provide integer clone sizes but rather area of clones, so we can make the transformation $n = \rho A$:

$$g_A(t) \approx \frac{\mu n_0}{r\lambda\rho A} e^{-\frac{\rho A}{N(t)}}$$

This is the result presented in the main text.

## Code and data availability

Code to reproduce all the figures in the manuscript (using a snakemake [*Köster and Rahmann, 2012*] workflow) is available at github.com/marcjwilliams1/dnds-clonesize (*Williams, 2020*; copy archived at https://elifesciences-publications/dnds-clonesize). We also created a singularity image with all software dependencies which is available at shub://marcjwilliams1/dnds-clonesize-R-container. Julia (*Bezanson et al., 2017*) was used for the simulations and R (*R Development Core Team, 2019*) was used to analyse the data and generate the figures. Some of the analysis rely in bespoke packages written for this study which are freely available under an open source licence.

## Acknowledgements

AS is supported by the Wellcome Trust (202778/B/16/Z) and Cancer Research UK (A22909). TG is supported by the Wellcome Trust (202778/Z/16/Z) and Cancer Research UK (A19771). We acknowledge funding from the National Institute of Health (NCI U54 CA217376) to AS and TG This work was also supported a Wellcome Trust award to the Centre for Evolution and Cancer (105104/Z/14/Z). CPB is supported by the Wellcome Trust (209409/Z/17/Z).

## Additional information

### Funding

| Funder | Grant reference number | Author |
| --- | --- | --- |
| Wellcome | 202778/B/16/Z | Andrea Sottoriva |
| Wellcome | 202778/Z/16/Z | Trevor A Graham |
| Wellcome | 209409/Z/17/Z | Chris P Barnes |
| Wellcome | 105104/Z/14/Z | Andrea Sottoriva |
| Cancer Research UK | A22909 | Andrea Sottoriva |
| Cancer Research UK | A19771 | Trevor A Graham |
| H2020 Marie Skłodowska-Curie Actions | 846614 | Luis Zapata |
| National Institutes of Health | CA217376 | Andrea Sottoriva Trevor A Graham |

The funders had no role in study design, data collection and interpretation, or the decision to submit the work for publication.

### Author contributions

Marc J Williams, Conceptualization, Data curation, Software, Formal analysis, Investigation, Visualization, Methodology; Luis Zapata, Benjamin Werner, Formal analysis, Investigation, Methodology; Chris P Barnes, Conceptualization, Software, Formal analysis, Supervision, Funding acquisition, Investigation, Methodology; Andrea Sottoriva, Conceptualization, Formal analysis, Supervision, Funding acquisition, Investigation, Methodology, Project administration; Trevor A Graham, Conceptualization, Resources, Formal analysis, Supervision, Funding acquisition, Investigation, Methodology, Project administration

### Author ORCIDs

Marc J Williams https://orcid.org/0000-0001-5524-4174
Luis Zapata https://orcid.org/0000-0002-1386-2019
Benjamin Werner https://orcid.org/0000-0002-6857-8699
Chris P Barnes http://orcid.org/0000-0002-9459-1395
Andrea Sottoriva https://orcid.org/0000-0001-6709-9533
Trevor A Graham https://orcid.org/0000-0001-9582-1597

### Decision letter and Author response

Decision letter https://doi.org/10.7554/eLife.48714.sa1
Author response https://doi.org/10.7554/eLife.48714.sa2

## Additional files

### Supplementary files

• Transparent reporting form

## Data availability

No new data was generated in this; only previously published data is reanalysed. Computer code implementing the new mathematical theory we developed is available here: https://github.com/marcjwilliams1/dnds-clonesize (copy archived at https://github.com/elifesciences-publications/dnds-clonesize).

The following previously published datasets were used:

| Author(s) | Year | Dataset title | Dataset URL | Database and Identifier |
|---|---|---|---|---|
| Martincorena I, Fowler JC Wabik A, Lawson AAR, Abascal F, Michael Hall WJ, Cagan A, Murai k, Mahbubani K, Stratton MR, Fitzgerald RC, Handford PA, Campbell PJ, Saeb-Parsy K, Jones PH | 2018 | EGAD00001004158 | https://www.ebi.ac.uk/ega/datasets/EGAD00001004158 | European Genome-phenome Archive, EGAD00001004158 |
| Martincorena I, Fowler JC Wabik A, Lawson AAR, Abascal F, Michael Hall WJ, Cagan A, Murai k, Mahbubani K, Stratton MR, Fitzgerald RC, Handford PA, Campbell PJ, Saeb-Parsy K, Jones PH | 2018 | EGAD00001004159 | https://www.ebi.ac.uk/ega/datasets/EGAD00001004159 | European Genome-phenome Archive, EGAD00001004159 |
| Martincorena I, Roshan A, Gerstung M, Ellis P, Van Loo P, McLaren S, Wedge DC, Fullam A, Alexandrov LB, Tubio JM, Stebbings L, Menzies A, Widaa S, Stratton MR, Jones PH, Campbell PJ | 2016 | EGAS00001000860 | https://www.ebi.ac.uk/ega/datasets/EGAS00001000860 | European Genome-phenome Archive, EGAS00001000860 |
| Martincorena I, Roshan A, Gerstung M, Ellis P, Van Loo P, McLaren S, Wedge DC, Fullam A, Alexandrov LB, Tubio JM, Stebbings L, Menzies A, Widaa S, Stratton MR, Jones PH, Campbell PJ | 2016 | EGAS00001000515 | https://www.ebi.ac.uk/ega/datasets/EGAS00001000515 | European Genome-phenome Archive, EGAS00001000515 |
| Martincorena I, Roshan A, Gerstung M, Ellis P, Van Loo P, McLaren S, Wedge DC, Fullam A, Alexandrov LB, Tubio JM, Stebbings L, Menzies A, Widaa S, Stratton MR, Jones PH, Campbell PJ | 2016 | EGAS00001000603 | https://www.ebi.ac.uk/ega/datasets/EGAS00001000603 | European Genome-phenome Archive, EGAS00001000603 |

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
