## [Decision Letter]

**Acceptance summary:**

The paper represents first a new contribution to the methodology to infer selection coefficients associated of mutations, and, second, the application of the methodology to the comparative study of mutations in normal versus cancer sub-clones. The general findings contribute to our understanding of somatic mutations and tumor evolution, as well as to the ongoing discussion on the role of potential cancer-related mutations in normal tissues.

**Decision letter after peer review:**

Thank you for submitting your article "Measuring the distribution of fitness effects in somatic evolution by combining clonal dynamics with dN/dS ratios" for consideration by *eLife*. Your article has been reviewed by Patricia Wittkopp as the Senior Editor, a Reviewing Editor, and three reviewers. The following individuals involved in review of your submission have agreed to reveal their identity: Ferran Muinos (Reviewer #1); Jamie Blundell (Reviewer #2).

The reviewers have discussed the reviews with one another and the Reviewing Editor has drafted this decision to help you prepare a revised submission.

Summary:

The paper deals with the important issue of understanding how selection of quantifying the selective advantage provided by somatic mutations as a key element to understand tumor evolution.

The proposed method infers selection coefficients associated with mutations that expand in normal tissues as well as subclonal mutations in oesophageal and skin data set (Martincorena et al.,) linking dN/dS and clone size and provides a model of somatic evolution in normal tissues, applying it to the and infer selective advantages for specific genetic loci in normal and cancer tissues, effectively linking SFS and dN/dS ratios..

The paper is timely, since a number of intriguing reports have detected somatic evolution (potential positive selection) in normal tissues that should not be pathogenic. It also aligns well with previous publications; bioRxiv 569566; doi: https://doi.org/10.1101/569566

Essential revisions:

The referees and editors truly appreciate the approach and consider it an important contribution to the field. Still, we find that it is important to validate the main hypothesis on the estimation of the selection coefficient on mutations in normal tissues.

We will also ask you to consider if the section on cancer is essential for the paper, as it is it is problematic in terms reproducibility and seems rather a distraction of the main argument.

During the revision, you will have the opportunity to address following issues related with clarifications and additional information. We find particularly important to address the ones related with the importance of different mutations in the same gene and the influence of history, e.g. influence of germline mutations in clonal evolution and the potentially transient nature of fitness gains.

Limited applicability questioning the application to cancer cases.

- The method for inferring selective advantages in tumors has one important limitation which is made very clear in the text: the method is not ready for the commonly available data sets. Remarkably, it is apparent that even in the case of high-quality histopathologically-normal tissue data the method has still limited resolution. I wonder whether this low resolution may compromise some of the conclusions drawn after the analysis.

- It is very reassuring from the modeling perspective that the model and subsequent fitting approach yields consistent results with the simulations when applied to the normal tissue data. However, the validation in tumors is lacking. Simulation analyses only confirm that the implementation of the fitting method correctly reflects the model, regardless of the validity of the model.

- The method heavily resorts to the estimation of mutation frequencies. However, this estimation may be difficult, particularly from tumor data with complex chromosomal architectures.

- In summary, there is not enough data for cancer. It is unclear if the section of application to cancer adds much, given the clear statement that there is not enough subclonal recurrence to say very much.

Different mutations in the same gene

- Although it may not affect the validity of the method, the conceptualization of driver mutation presented in the manuscript may obfuscate the interpretation of the results. It is very likely that any mutation found in a gene subject to high selective pressure confers a selective advantage itself, but for genes under mild positive selection this may not be true, whence not all mutations may be bound to drive. Moreover, different driver mutations may confer distinct selective advantages.

- A challenge in using the skin / esophagus data is that the lack of large numbers of people sampled means data is rather sparse. Our experience looking at the blood makes clear that there is often a wide spectrum of fitness effects even for non-syn mutations within the same gene. This effect is not discussed here. Can the authors present an analysis of say the bottom few genes in Figure 3A to test whether there is evidence that the Δ are the same or whether some of the wide variation could be attributable to a variation of Δ even within the gene? I appreciate the data might not be there for a thorough analysis of this though.

- To easily track all the assumptions implied by the models it would be very beneficial for the manuscript to include sketches of the proofs that formally derive the two main functional forms adopted for the mutation density g. This would make the line of reasoning much more transparent and spare the reader the time for literature scrapping.

- Although from the text it seems that the variants implied under the term "mutation" are just coding SNVs, the analysis may well be valid for other types of mutations such as small insertions and deletions or splicing affecting SNVs, as the dN/dS method of choice is equipped to handle them. Please, clarify. Moreover, dNdScv is dependent on the choice of a transcript for the genes analyzed: this choice should be pointed out somewhere in the manuscript.

- The method heavily resorts to the availability of a sufficiently robust method to estimate dN/dS at genetic loci from mutational data. Thus, all the caveats of the method of choice may propagate downstream onto the method described, it is good that these limitations are discussed in the text.

- Motivated by (Introduction) "Enumeration of the selective advantage of each driver mutation enables prediction of future evolutionary dynamics". This claim is more difficult to interpret in the light of your own findings, namely, that there are somatic mutations showing a very significant fitness gain in normal tissues. Consider that those mutations may confer a transient fitness gain until a new homeostatic equilibrium is attained.

- How are the step-sizes of the clone size intervals decided? How coarse these bins can be while still rendering a meaningful fit for the selective coefficients?

- The Discussion section is mainly descriptive and methodological. What can be concluded about the role of NOTCH1 and TP53 in aging and/or tumor initiation in the light of the results in normal tissues?

- The methods are essentially the same (e.g. Equation 4 in this work is Equation 1 in Watson et al., except Watson et al., focusses on SFS view whereas Williams et al. focus on the frequency-dependent dN/dS view – though, as the authors know well, these are different ways of viewing exactly the same info!). It should be made very clear that this has been independently developed by Watson et al. This is especially true since the way Watson et al., is cited at the moment is wholly incorrect!

- Internal consistency between syn and non-syn mutations and validating N. The framework presented here, at its heart, analyses the SFS of two classes of mutations (syn vs non-syn) and takes the ratio of these in bins of increasing frequency. However, taking the ratio of the SFS for these classes of mutation throws away important information and means the authors lose the ability to validate N. If instead of taking the ratios they simply plot the SFS for both the syn and nonsyn variants independently then the amplitude of these curves at low frequency will be given by Nμ, which given one can estimate μ enables one to calculate N, which serves as an important validation since you can check agreement with known estimates of 5000/mm2 (as quoted in the text). I would strongly suggest plotting the SFS of each class of mutation independently as outlined in Watson et al., since this does not throw out the N dependence. Furthermore, this inference of N from the amplitude provides an important internal consistency check on characteristic frequency of the clones (what the authors term N(t) in their main text – although as outlined at the end of my review there is a minor error in their equation). The value of N enables one to predict where the synonynous variants should appear as outlined in (Watson et al. bioRxiv 569566; doi: https://doi.org/10.1101/569566). This analysis would greatly strengthen this work as it would challenge the model more substantially.

- Validation of Δ. One major flaw of the work as presented is that the inferences for Δ are not validated or challenged. One approach we found useful is that the Δ make clear predictions for how the SFS should evolve with age. Even though there is no longitudinal sampling, there is a very wide range of ages in the esophagus data and reasonably wide range in the skin data. The authors should check whether the Δ inferred is consistent with how the SFS changes with age. i.e there should be a wider spectrum of frequencies in the older people and a narrower spectrum in younger people. In fact, the characteristic frequency of the clones should increase exponentially with age since the N(t) (characteristic size of the extant non-syn clones) increases exponentially with age so this makes a clear prediction to be checked. Including this analysis would strengthen the manuscript considerably.

- Spatial effects. A challenge with using data from skin and esophagus to check this theoretical framework is that these tissues evolve on a 2D spatially constrained sheet of cells. There was no development or real discussion of how this might affect the theory (which has no explicit spatial effects built in). While a full theoretical treatment of this does seem beyond the scope of this work, it would be nice to see a thorough discussion of it, especially relating to some of the findings in the recent literature e.g. Hall et al., bioRxiv 480756; doi: https://doi.org/10.1101/480756 and Fusco et al., 2016, neither of which were cited.

- High frequency synonymous variants are likely to be genetic hitchhikers. While there was a short discussion of hitchhikers at the end, it was somewhat cursory and not quantitative. Given the theory, the authors can predict what fraction of syn variants are expected to be hitchhikers in the skin / esophagus data and their frequency distribution (this analysis has been performed in Supplementary note 7 of Watson et al., 2019). It would be an important check on the inferences to perform a similar analysis and see if there is agreement with predictions.

[Editors' note: further revisions were suggested prior to acceptance, as described below.]

Thank you for resubmitting your work entitled "Measuring the distribution of fitness effects in somatic evolution by combining clonal dynamics with dN/dS ratios" for further consideration by *eLife*. Your revised article has been evaluated by Patricia Wittkopp (Senior Editor), a Reviewing Editor, and the original reviewers.

The manuscript has been improved but the two referees still have a number of points that require additional clarification or a better explanation. The one that I find compulsory is a better description of Figure 5, as detailed by the second referee.

*Reviewer #1:*

- From this reviewer's perspective, the main concerns raised during the first round of reviews have been addressed. The authors have conceded to reshape the paper so as to exclude the cancer part in the interest of applicability and reproducibility.

- The authors have thoroughly addressed all the comments including additional explanations and new figures for the sake of completeness and clarification. In particular, the authors have now conducted a supplementary analysis of the impact of the dN/dS calculation method of choice; have completed the methodological description; have added clarification to the relationship between predictiveness and selective regime; have completed the discussion hypothesizing a temporal dynamics for the expansion of NOTCH1 vs TP53 variants in histopathologically normal oesophageal tissue; have tackled the relevance and limitations of the study regarding the per-site inference of selective advantages; have clarified the impact of bin-sizes at model fitting; and have carried an important new analysis suggested by one of the reviewers to validate Δ.

- The manuscript has clearly improved and feels to have reached a point so that exposure to public scientific debate cannot wait anymore.

*Reviewer #2:*

Generally I find the manuscript improved relative to the first draft. However there remain a few issues that I believe need to be addressed to really demonstrate whether the numbers inferred using this method are meaningful.

I am glad to see the suggestion made in my first review on ways to challenge the inferred Δ was used by the authors as the basis for the substantial new analysis resulting in Figure 5. I believe the additional analysis strengthens the manuscript. Given that the idea for this analysis comes from an analysis performed in Watson et al., it would be appropriate to clearly acknowledge this in the main text e.g. words to the effect of "Following closely the analysis laid out in Watson et al. we reasoned…" subsection “Site frequency spectra”.

I also have a few questions about this current analysis. First, as mentioned by the authors, the theoretical prediction is that N(t) should increase exponentially with age with the gradient being approximately Δ. In Figure 5D it is not clear to me what data has been used for this plot and how numbers were estimated. If this data is grouped across many genes the predicted increase should not be exponential at all, so it is not clear if this fit is appropriate. The authors need to clarify this. It is more appropriate to perform the analysis on a gene-by-gene basis as shown in Figure 5—figure supplement 1 but this figure highlights that the agreement between the values of Δ using the two methods are not very good (e.g. TP53 is inferred to be ~0.018 on basis of age change vs 0.06 on basis of dN/dS). The log scale on Figure 5—figure supplement 1B obfuscates this. Is the correlation between the methods still significant on a (more standard) linear scale? The y-axis scale on Figure 5C is incorrect: there is a missing factor of rλ (see below too). Generally I think this new analysis is encouraging but would benefit from a more cautionary telling: highlighting how the agreement between inferred values of Δ is far from perfect and might point to more complex dynamics e.g. environmental effects which alter Δ.

There remains an error in Equation. 4. The units are not correct. The density is dimensionless while at present it is proportional to mutation rate (units of 1/time). There is a timescale missing from the expression: which I believe is r \λ and should be in the denominator. The authors should check this.

I am still not convinced the authors have satisfactorily addressed the issue of sites within a gene having different Δ values. While I accept that the current data sets do not have the power to resolve Δ on a site by site basis they can, and in my opinion should, simulate this with some parameterised distribution and test what value of Δ they would infer using their grouped method. Because, if there is a range of Δ for mutations within the same gene, and a single value is inferred, it is not clear that the value inferred represents the average value of Δ across the gene as the authors claim in their rebuttal. This claim should be backed up with simulations if the data do not exist.

---

## [Author Response]

Summary:The paper deals with the important issue of understanding how selection of quantifying the selective advantage provided by somatic mutations as a key element to understand tumor evolution.The proposed method infers selection coefficients associated with mutations that expand in normal tissues as well as subclonal mutations in oesophageal and skin data set (Martincorena et al.,) linking dN/dS and clone size and provides a model of somatic evolution in normal tissues, applying it to the and infer selective advantages for specific genetic loci in normal and cancer tissues, effectively linking SFS and dN/dS ratios..The paper is timely, since a number of intriguing reports have detected somatic evolution (potential positive selection) in normal tissues that should not be pathogenic. It also aligns well with previous publications; bioRxiv 569566; doi: https://doi.org/10.1101/569566

We thank the reviewers and editor for this accurate summary and their constructive and thorough critique of our study. We are glad that our paper is judged important and timely.

Essential revisions:The referees and editors truly appreciate the approach and consider it an important contribution to the field. Still, we find that it is important to validate the main hypothesis on the estimation of the selection coefficient on mutations in normal tissues.We will also ask you to consider if the section on cancer is essential for the paper, as it is it is problematic in terms reproducibility and seems rather a distraction of the main argument.During the revision, you will have the opportunity to address following issues related with clarifications and additional information. We find particularly important to address the ones related with the importance of different mutations in the same gene and the influence of history, e.g. influence of germline mutations in clonal evolution and the potentially transient nature of fitness gains.

We are glad that the editor and reviewers judge our study favourably. Below, we have addressed the various specific points that challenge (and thus serve to help verify) the main findings of the paper. We have also attempted to refine our analysis as suggested, wherever was feasible to do so. Major changes to the manuscript include validation of the selection coefficients by directly fitting the site frequency spectrum, implementing an alternative dN/dS method and checking that the choice of bin size has limited influence on inferred selection coefficients. Details are provided below.

We have now restricted the manuscript and removed the section on cancer. We agree that the lack of sufficient data in cancer diminishes interest in that section of the manuscript. We would, of course, intend to revisit this as sufficient data become available.

Limited applicability questioning the application to cancer cases.- The method for inferring selective advantages in tumors has one important limitation which is made very clear in the text: the method is not ready for the commonly available data sets. Remarkably, it is apparent that even in the case of high-quality histopathologically-normal tissue data the method has still limited resolution. I wonder whether this low resolution may compromise some of the conclusions drawn after the analysis.- It is very reassuring from the modeling perspective that the model and subsequent fitting approach yields consistent results with the simulations when applied to the normal tissue data. However, the validation in tumors is lacking. Simulation analyses only confirm that the implementation of the fitting method correctly reflects the model, regardless of the validity of the model.- The method heavily resorts to the estimation of mutation frequencies. However, this estimation may be difficult, particularly from tumor data with complex chromosomal architectures.- In summary, there is not enough data for cancer. It is unclear if the section of application to cancer adds much, given the clear statement that there is not enough subclonal recurrence to say very much.

We agree with this assessment and have now removed the cancer section from the paper.

We also agree that accurate determination of mutant allele frequency is much more problematic in tumour data compared to the ostensibly normal tissue data from Martincorena et al., largely due to copy number alterations and tumour purity. This is a major reason why the cancer data, despite the large number of samples, cannot be analysed with our method.

Different mutations in the same gene- Although it may not affect the validity of the method, the conceptualization of driver mutation presented in the manuscript may obfuscate the interpretation of the results. It is very likely that any mutation found in a gene subject to high selective pressure confers a selective advantage itself, but for genes under mild positive selection this may not be true, whence not all mutations may be bound to drive. Moreover, different driver mutations may confer distinct selective advantages.- A challenge in using the skin / esophagus data is that the lack of large numbers of people sampled means data is rather sparse. Our experience looking at the blood makes clear that there is often a wide spectrum of fitness effects even for non-syn mutations within the same gene. This effect is not discussed here. Can the authors present an analysis of say the bottom few genes in Figure 3A to test whether there is evidence that the Δ are the same or whether some of the wide variation could be attributable to a variation of Δ even within the gene? I appreciate the data might not be there for a thorough analysis of this though.

We entirely agree that different mutations in the same gene may have different selective advantages. Indeed, this hypothesis partially explains the phenomenon of ‘hotspot’ mutations (mutations always in the same site(s)) in cancer-associated genes. Consequently, we recognize that a selection coefficient per site would be the most informative way to characterize selection in somatic genomes.

While in theory the approach we have developed could be applied at a site-specific level, we are limited by both the data at hand and methods to evaluate dN/dS at the site level. In terms of the data we used for this study, in some donors in the oesophagus dataset the same mutation appears to arise multiple times independently, but the frequency of these events is always below the 8 mutations necessary to apply our approach (maximum 6 repeated occurrences of the same mutation observed). The reviewer will appreciate that normalising for underlying mutation rate at a site by site level (as opposed to the average rate across a larger locus) is also problematic.

Unfortunately, we are therefore unable to assess selection coefficients at the site level in the available data. We have added sentences to the main text making clear that the inferences we draw are effectively average selection coefficients across the whole gene. We also discuss in more detail how this approach could be extended to the site level with larger cohorts and improved methods to estimate site level dN/dS (Discussion section).

- To easily track all the assumptions implied by the models it would be very beneficial for the manuscript to include sketches of the proofs that formally derive the two main functional forms adopted for the mutation density g. This would make the line of reasoning much more transparent and spare the reader the time for literature scrapping.

We have now included more detail on the proofs at the end of the Materials and methods section.

- Although from the text it seems that the variants implied under the term "mutation" are just coding SNVs, the analysis may well be valid for other types of mutations such as small insertions and deletions or splicing affecting SNVs, as the dN/dS method of choice is equipped to handle them. Please, clarify. Moreover, dNdScv is dependent on the choice of a transcript for the genes analyzed: this choice should be pointed out somewhere in the manuscript.

We have now included more details on the mutations used and the references used to annotate transcript in the subsection “dN/dS calculations”. The reviewer is correct that in principle any type of mutation can be used for the analysis, provided that there is a good null model for the underlying variation in mutation rate across sites/loci.

- The method heavily resorts to the availability of a sufficiently robust method to estimate dN/dS at genetic loci from mutational data. Thus, all the caveats of the method of choice may propagate downstream onto the method described, it is good that these limitations are discussed in the text.

This is an important point, and we are grateful to the reviewers for raising it. To examine the impact of dN/dS calculation methodology, we have now compared the results using a distinct dN/dS method, called SSB-dNdS from Zapataet al., (10.1186/s13059-018-1434-0). This showed consistent results with dndscv. Specifically, we observed the same shape curve in the analysis (Figure 5—figure supplement 1A) and also very consistent results for global dN/dS values for each patient (Figure 5—figure supplement 1C). Gene level values show some variability between the methods, likely due differences between the two methodologies in normalising for local variations in the mutation rate (Figure 5—figure supplement 1B).

We now discuss this issue at greater length (Materials and methods section).

- Motivated by (Introduction) "Enumeration of the selective advantage of each driver mutation enables prediction of future evolutionary dynamics". This claim is more difficult to interpret in the light of your own findings, namely, that there are somatic mutations showing a very significant fitness gain in normal tissues. Consider that those mutations may confer a transient fitness gain until a new homeostatic equilibrium is attained.

We agree that selective regimes can change over time, due to changes in the microenvironmental context (indeed, in the case that the reviewer suggests it is the changing degree of competition with neighbouring cells). Nevertheless, we maintain that quantitative knowledge of the evolutionary dynamics allows for predictions about the future, but that these predictions are only valid for the duration of the selective regime where the measurement was made. We have added this caveat (Introduction).

- How are the step-sizes of the clone size intervals decided? How coarse these bins can be while still rendering a meaningful fit for the selective coefficients?

We used a binsize of 0.025 (area of clone in mm^2^) for our inferences, however the size of bins makes little difference to the inferences as we integrate over bins in our interval-dN/dS method. Indeed, this is one of the reasons we fitted our model using the integral, as this avoids the trade-offs between binsize and the number of data points in each bin.

To confirm that the binsize has little effect we generated a dataset with our simulation with a known ground truth of ∆= 0.4 and fitted the data with a range of bin sizes, these results showed consistent estimates for ∆ but with progressively larger confidence intervals at larger bin sizes (Author response image 1).

**Author response image 1. respfig1:** Comparison of different bin sizes. (**a**) Fits to simulation with different bin sizes (**b**) Summary of inferred ∆ for the different bin sizes shows highly consistent results with a small increase in variance at larger bin sizes.

- The Discussion section is mainly descriptive and methodological. What can be concluded about the role of NOTCH1 and TP53 in aging and/or tumor initiation in the light of the results in normal tissues?

We have now broadly revised the discussion following the removal of the cancer section. Following the reviewers’ suggestion, we have also now included a discussion of the hypothesised temporal dynamics of NOTCH1 and TP53 mutation accumulation: we suppose that mutationally-likely NOTCH1-mutants initially repopulate the oesophagus before being replaced by the later arising (mutationally less likely) but fitter TP53-mutant clones (Discussion section).

- The methods are essentially the same (e.g. Equation 4 in this work is Equation 1 in Watson et al., except Watson et al., focusses on SFS view whereas Williams et al. focus on the frequency-dependent dN/dS view – though, as the authors know well, these are different ways of viewing exactly the same info!). It should be made very clear that this has been independently developed by Watson et al. This is especially true since the way Watson et al., is cited at the moment is wholly incorrect!

We apologise for mis-citing the Watson et al., study, it is now referred to appropriately (Introduction).

We feel that here it is important to note that the mathematical foundations of our study and that of Watson et al., have long routes in the literature (indeed, the fundamentals go back to Luria and Delbruck and more general classical branching process theory). Moreover, the application of these mathematical ideas to problems in biology is also a well-trodden path, with the large body of work from Benjamin Simons over the past decade being exemplary of this. Thus, with respect, we do not think it appropriate to attribute the mathematical foundations of our study to Watson and colleagues’ paper. We have explained the background of the ‘clonal dynamics inference’ approach in general and in biology in the revised main manuscript (Introduction and Materials and methods section).

- Internal consistency between syn and non-syn mutations and validating N. The framework presented here, at its heart, analyses the SFS of two classes of mutations (syn vs non-syn) and takes the ratio of these in bins of increasing frequency. However, taking the ratio of the SFS for these classes of mutation throws away important information and means the authors lose the ability to validate N. If instead of taking the ratios they simply plot the SFS for both the syn and nonsyn variants independently then the amplitude of these curves at low frequency will be given by Nμ, which given one can estimate μ enables one to calculate N, which serves as an important validation since you can check agreement with known estimates of 5000/mm2 (as quoted in the text). I would strongly suggest plotting the SFS of each class of mutation independently as outlined in Watson et al., since this does not throw out the N dependence. Furthermore, this inference of N from the amplitude provides an important internal consistency check on characteristic frequency of the clones (what the authors term N(t) in their main text – although as outlined at the end of my review there is a minor error in their equation). The value of N enables one to predict where the synonynous variants should appear as outlined in (Watson et al., bioRxiv 569566; doi: https://doi.org/10.1101/569566). This analysis would greatly strengthen this work as it would challenge the model more substantially.

We are grateful for this suggestion, and have considered it in some detail. However, we do not think that the approach is feasible in our current study. Watson et al., used an orthogonal dataset (whole genome sequencing of multiple samples from a single individual) to estimate the mutation rate, which they could then use to decouple the mutation rate from the number of stem cells. However, we do not think that there exists a suitable dataset derived from oesophagus or skin where we could do similar.

Alternatively, we could try and estimate the mutation rate from the Martincorena data itself, however there is the obvious risk of making a circular argument. These data are targeted sequencing of a panel of 72 genes. Our analysis, and indeed the original study too, demonstrate that many of these genes are under strong positive selection, rendering inference of mutation rates problematic.

We do however recognize that directly fitting the site frequency spectrum can provide additional ways to validate our approach, we discuss this in more detail in the next point.

- Validation of Δ. One major flaw of the work as presented is that the inferences for Δ are not validated or challenged. One approach we found useful is that the Δ make clear predictions for how the SFS should evolve with age. Even though there is no longitudinal sampling, there is a very wide range of ages in the esophagus data and reasonably wide range in the skin data. The authors should check whether the Δ inferred is consistent with how the SFS changes with age. i.e there should be a wider spectrum of frequencies in the older people and a narrower spectrum in younger people. In fact, the characteristic frequency of the clones should increase exponentially with age since the N(t) (characteristic size of the extant non-syn clones) increases exponentially with age so this makes a clear prediction to be checked. Including this analysis would strengthen the manuscript considerably.

This is an excellent point and we thank the reviewer for highlighting the additional information provided by the site frequency spectrum and the ways it could be used to validate the results from our interval-dN/dS approach. We have now added a new Figure 5 where we analyse the site frequency spectrum directly to attempt to validate our interval-dN/dS derived conclusions.

Analysis of the SFS reveals the age-dependent trends that the reviewer expects, namely larger clones and greater variability in clone size in older people (Figure 5A). We also observe the prediction that the characteristic frequency of non-synonymous mutations should increase markedly with age and a more modest increase for synonymous mutations (Figure 5D).

We also now challenge the underlying theoretical model of clonal expansion dynamics by fitting two alternate models of clonal dynamics directly to the site frequency spectrum. The three models we fit are as follows (y is clone size and A and B are parameters to be inferred):

1) y~Ane-nB

2) y~An

3) y~Ae-nB

Here model 1 is the “correct” model and models 2 and 3 are simplifications which only include the exponential decay or a power law. Using the widely applicable information criteria (WAIC), we find that the ‘correct’ model 1 provides the best predictive power (lowest WAIC) (Figure 5B).

We also performed the fit of the clonal dynamics model on a gene-by-gene basis, however due to limited data points we found the uncertainty in our posterior estimates of the parameters to be large, precluding further insight at the gene level with this approach. Figure 5—figure supplement 3 shows the fits for *NOTCH1* and *TP53* and posterior estimates of the characteristic frequency.

As an alternative method to ‘validate’ our gene-level selective coefficients, we therefore turned to a purely statistical model and performed a Bayesian multi-level regression on area~age with genes as random effects. This analysis found that *TP53* and *NOTCH1* were the genes with the highest coefficients for the slopes (ie mutations in genes increase in clone size as a function of age to a greater extent than all other genes (Figure 5—figure supplement 5A)). Further, these regression coefficients were correlated with our inferred selection coefficients, supporting the conclusions from our interval dN/dS method (Figure 5—figure supplement 4B).

In summary, direct analysis of the site frequency spectrum data support our original conclusions.

- Spatial effects. A challenge with using data from skin and esophagus to check this theoretical framework is that these tissues evolve on a 2D spatially constrained sheet of cells. There was no development or real discussion of how this might affect the theory (which has no explicit spatial effects built in). While a full theoretical treatment of this does seem beyond the scope of this work, it would be nice to see a thorough discussion of it, especially relating to some of the findings in the recent literature e.g. Hall et al., bioRxiv 480756; doi: https://doi.org/10.1101/480756 and Fusco et al., 2016, neither of which were cited.

We agree that spatial effects can be important, we have added a note about this and cited both papers in the Discussion section. We have assumed that clones experience minimal interference from other clones and so spatial effects are minimised. There is some supporting evidence for the reasonableness of this assumption that is provided below in response to the hitchhiker comment.

- High frequency synonymous variants are likely to be genetic hitchhikers. While there was a short discussion of hitchhikers at the end, it was somewhat cursory and not quantitative. Given the theory, the authors can predict what fraction of syn variants are expected to be hitchhikers in the skin / esophagus data and their frequency distribution (this analysis has been performed in Supplementary note 7 of Watson et al., 2019). It would be an important check on the inferences to perform a similar analysis and see if there is agreement with predictions.

To look at the influence of hitchhikers in the oesophagus dataset we leveraged the spatial information in the sequenced patches. In the dataset each donor had approximately 90 distinct small pieces of tissue that were sequenced, and we note that each of these pieces contained multiple clones. We reasoned that pieces containing selected clones would be expected to have a larger number of detectable synonymous mutations if hitchhiking was common. We also expect the hitchhiking synonymous mutations to have larger clone size than non-hitchhikers. To test these hypotheses we counted the number of non-synonymous *NOTCH1* and *TP53* mutations (these being the clones with the highest fitness) as well as the number of synonymous mutations (and their VAF) in each piece of tissue.

Figure 5—figure supplement 1A,B shows the number of synonymous mutations as a function of the number of nonsynonymous *NOTCH1* and *TP53* mutations in a patch. Figure 5—figure supplement 1C,D shows the VAF of synonymous mutations as a function of the number of non-synonymous NOTCH1 and TP53 mutations in a patch. There was no relationship between the number of selected clones and synonymous mutation VAF (by linear regression). A significant correlation between the number of selected clones and synonymous mutation burden was observed (p=0.0002 for NOTCH1, p=0.031 for TP53) suggesting the presence of a small number of synonymous hitchhikers in the dataset. The regression coefficients show that for each additional nonsynonymous mutation we get ~0.05 (0.047 for NOTCH1 and 0.057 for TP53) additional synonymous mutations. We note too that the correlation is very noisy (R^2^<0.02). Thus, we conclude that the number of hitchhiking synonymous mutations is minimal.

[Editors' note: further revisions were suggested prior to acceptance, as described below.]

Reviewer #1:- From this reviewer's perspective, the main concerns raised during the first round of reviews have been addressed. The authors have conceded to reshape the paper so as to exclude the cancer part in the interest of applicability and reproducibility.- The authors have thoroughly addressed all the comments including additional explanations and new figures for the sake of completeness and clarification. In particular, the authors have now conducted a supplementary analysis of the impact of the dN/dS calculation method of choice; have completed the methodological description; have added clarification to the relationship between predictiveness and selective regime; have completed the discussion hypothesizing a temporal dynamics for the expansion of NOTCH1 vs TP53 variants in histopathologically normal oesophageal tissue; have tackled the relevance and limitations of the study regarding the per-site inference of selective advantages; have clarified the impact of bin-sizes at model fitting; and have carried an important new analysis suggested by one of the reviewers to validate Δ.- The manuscript has clearly improved and feels to have reached a point so that exposure to public scientific debate cannot wait anymore.

We thank the reviewer for the positive assessment of our manuscript.

Reviewer #2:Generally I find the manuscript improved relative to the first draft. However there remain a few issues that I believe need to be addressed to really demonstrate whether the numbers inferred using this method are meaningful.I am glad to see the suggestion made in my first review on ways to challenge the inferred Δ was used by the authors as the basis for the substantial new analysis resulting in Figure 5. I believe the additional analysis strengthens the manuscript. Given that the idea for this analysis comes from an analysis performed in Watson et al. it would be appropriate to clearly acknowledge this in the main text e.g. words to the effect of "Following closely the analysis laid out in Watson et al. we reasoned…" subsection “Site frequency spectra”.

We thank the reviewer for recognizing that the new analysis strengthens the manuscript and for motivating us to perform this analysis. We have added the reference to Watson et al., and also to the other earlier papers that perform a similar analysis.

I also have a few questions about this current analysis. First, as mentioned by the authors, the theoretical prediction is that N(t) should increase exponentially with age with the gradient being approximately Δ. In Figure 5D it is not clear to me what data has been used for this plot and how numbers were estimated. If this data is grouped across many genes the predicted increase should not be exponential at all, so it is not clear if this fit is appropriate.

Yes, the analysis is performed on all genes. To assess whether this is a reasonable approach we generated a synthetic cohort where Δ was drawn from an exponential distribution with mean = 0.05. We observe the same trend in these simulations as we do in the real data. Author response image 2 is equivalent to Figure 5D. Comparison of the inferred values versus the ground truth (setting Δ = 0.05, the mean of the DFE), shows that our inferred values broadly agree, with some discrepancy at large times, Author response image 2. This gives us confidence that performing this analysis is appropriate and measures average effects across genes.

**Author response image 2. respfig2:** Fitted parameter values of a simulated dataset as a function of time. (**a**) Inferred N(t) as a function for age in a simulated cohort, equivalent to Figure 5D. (**b**) Posterior estimates for inferred values in black, with ground truth shown with red circular points, for this “ground truth” we calculated N(t) using Δ=0.05, the mean of the exponential distribution used in the simulations..

We also note that a large proportion of the non-synonymous mutations are in NOTCH1 and TP53 meaning that these genes are likely responsible for a large proportion of the effect we observe. Out of 813 nonsense mutations 36% (300) are in the NOTCH1 gene and 7% (59) are in TP53, of the 3558 missense mutations 27% (951) are in NOTCH1 and 11% (392) are in TP53.

The authors need to clarify this. It is more appropriate to perform the analysis on a gene-by-gene basis as shown in Figure 5—figure supplement 1 but this figure highlights that the agreement between the values of Δ using the two methods are not very good (e.g. TP53 is inferred to be ~0.018 on basis of age change vs 0.06 on basis of dN/dS). The log scale on Figure 5—figure supplement 1B obfuscates this. Is the correlation between the methods still significant on a (more standard) linear scale? The y-axis scale on Figure 5C is incorrect: there is a missing factor of rλ (see below too). Generally, I think this new analysis is encouraging but would benefit from a more cautionary telling: highlighting how the agreement between inferred values of Δ is far from perfect and might point to more complex dynamics e.g. environmental effects which alter Δ.

We apologise if some of the details of our approach were unclear, we have made some changes to the manuscript that we hope highlights the following points.

Firstly, we highlight that we attempted to fit the distribution per gene but found there were too few data points per gene which resulted in large (3 orders of magnitude) posterior estimates of parameters, see Figure 5—figure supplement 2.

As we could not fit the distributions per gene directly we resorted to a statistical model to assess which genes showed the largest increases in clone size as a function of age, reasoning that this would be a qualitative rather than quantitative validation of the fitness estimates (Figure 5—figure supplement 1). We have added this qualifier in subsection “Site frequency spectra” (“…providing qualitative support for our approach”). We therefore note that we would not expect the regression coefficients to match those values we infer from our i-dN/dS method. We do however observe that those genes with the largest fitness effects also have the largest regression coefficients, in line with predictions (in other words the bias is systematic, not random, and therefore we think the regression approach is provides useful insight).

That the regression model we used does not capture the underlying dynamics is nicely demonstrated through a posterior predictive check where we draw samples from the model fit and compare to data. As shown in Author response image 3 there is significant discrepancy between data and model (data is more left skewed) highlighting that this type of statistical model does not capture all of the dynamics.

**Author response image 3. respfig3:** Posterior predictive check of statistical model shown in Figure 5—figure supplementary 1. Dark blue density labelled y is the real data and lighter blue lines labelled yrep are datasets simulated from the posterior.

We note that one option to attempt to validate Δ is noting that the mean clone size, x as a function of age is given by (see Klein et al., equation S12 for derivation):

x=Nt-1log⁡(Nt)

However, due to the wide distribution of clone sizes, and limited number of patients there is considerable uncertainty with using this approach, as illustrated in Author response image 4.

**Author response image 4. respfig4:** Mean clone size as a function of age for NOTCH1 and TP53. Points represent the mean values and lines show 95% intervals.

In summary therefore, we believe this analysis serves as qualitative rather than quantitative support for our approach.

We have also changed the plot to be on a linear scale, the statistics we reported originally were on the linear scale so remain valid.

There remains an error in Equation. 4. The units are not correct. The density is dimensionless while at present it is proportional to mutation rate (units of 1/time). There is a timescale missing from the expression: which I believe is r \λ and should be in the denominator. The authors should check this.

We previously defined the mutation rate as mutations per symmetric division which is equivalent to the mutation rate per division scaled by the turnover rate, 𝑟𝜆. This definition was buried in the methods however, in order to clarify we now use 𝜇 for mutations per division and therefore include 𝑟𝜆 in the denominator in all relevant equations. We thank the reviewer for highlighting this potential source of confusion. We have therefore changed the Materials and methods section to reflect this.

I am still not convinced the authors have satisfactorily addressed the issue of sites within a gene having different Δ values. While I accept that the current data sets do not have the power to resolve Δ on a site by site basis they can, and in my opinion should, simulate this with some parameterised distribution and test what value of Δ they would infer using their grouped method. Because, if there is a range of Δ for mutations within the same gene, and a single value is inferred, it is not clear that the value inferred represents the average value of Δ across the gene as the authors claim in their rebuttal. This claim should be backed up with simulations if the data do not exist.

Using the same approach, we described above we have now performed these suggested simulations. The simulations provide confirmation that if there is a range of Δ for mutations within the same gene then the inferred Δ is an average effect for that gene.

To perform these simulations we assumed Δ was exponentially distributed, and generated 500 synthetic cohorts. Every new mutation entering the population in the simulations has an exponentially distributed fitness effect, and thus each cohort has a range of different fitness effects. We applied our method to estimate Δ for every cohort and assessed the values we estimated. Below is a histogram of these values for 3 different mean values of the exponential distribution, demonstrating that the value we estimate is consistent with the mean of the distribution of Δ. Median values and 95% intervals of these histograms are 0.078 (0.014 – 0.338), 0.114 (0.0224 – 0.382), 0.203 (0.0643 – 0.503). We have added Figure 2 —figure supplement 1 and included a description of these results in lines the Results section.